# BOTS: Batch Bayesian Optimization of Extended Thompson Sampling for Severely Episode-Limited RL Settings

**Karine Karine**
University of Massachusetts Amherst, USA
`karine@cs.umass.edu`

**Susan A. Murphy**
Harvard University, USA
`samurphy@g.harvard.edu`

**Benjamin M. Marlin**
University of Massachusetts Amherst, USA
`marlin@cs.umass.edu`

## Abstract

In settings where the application of reinforcement learning (RL) requires running real-world trials, including the optimization of adaptive health interventions, the number of episodes available for learning can be severely limited due to cost or time constraints. In this setting, the bias-variance trade-off of contextual bandit methods can be significantly better than that of more complex full RL methods. However, Thompson sampling bandits are limited to selecting actions based on distributions of immediate rewards. In this paper, we extend the linear Thompson sampling bandit to select actions based on a state-action utility function consisting of the Thompson sampler's estimate of the expected immediate reward combined with an action bias term. We use batch Bayesian optimization over episodes to learn the action bias terms with the goal of maximizing the expected return of the extended Thompson sampler. The proposed approach is able to learn optimal policies for a strictly broader class of Markov decision processes (MDPs) than standard Thompson sampling. Using an adaptive intervention simulation environment that captures key aspects of behavioral dynamics, we show that the proposed method can significantly out-perform standard Thompson sampling in terms of total return, while requiring significantly fewer episodes than standard value function and policy gradient methods.

## 1 Introduction

There is an increasing interest in using reinforcement learning methods (RL) in the healthcare setting, including in mobile health Coronato et al. [2020], Yu et al. [2021], Liao et al. [2022]. However, the healthcare domain presents a range of challenges for existing RL methods. In mobile health, each RL episode typically corresponds to a human subjects trial involving one or more participants that requires substantial time to carry out (weeks to months) and can incur significant cost. As a result, methods that require many episodes are usually not feasible [Williams, 1987, Mnih et al., 2013].

Within the mobile health research community specifically, adaptive intervention policy learning methods have addressed severe episode count restrictions imposed by real-world research constraints by focusing on the use of contextual bandit algorithms [Tewari and Murphy, 2017]. By focusing on maximizing immediate reward, bandit algorithms have the potential to provide an improved bias-variance trade-off compared to policy gradient and state-action value function approaches [Lattimore

Workshop on Bayesian Decision-making and Uncertainty, 38th Conference on Neural Information Processing Systems (NeurIPS 2024).

and Szepesvari, 2017]. Linear Thompson sampling (TS) bandits are a particularly promising approach due to the application of Bayesian inference to capture model uncertainty due to data scarcity in the low episode count setting [Agrawal and Goyal, 2013].

Of course, the main drawback of bandit-like algorithms is that they select actions based on distributions of immediate rewards, thus they do not account for long term consequences of present actions [Chapelle and Li, 2011]. This can lead to sub-optimal performance in real world applications where the environment corresponds to an arbitrary MDP, referred to as the "full RL" setting.

In this paper, we propose an approach to extending the linear TS bandit such that actions are selected using a state-action utility function that includes an action bias term learned across episodes using Bayesian Optimization (BO) applied to the expected return of the extended Thompson sampler. This approach retains much of the bias-variance trade-off of the classical TS bandit while having the ability to provide good performance for a strictly larger set of MDPs than the classical TS bandit.

Further, we explore the use of batch Bayesian optimization methods, including local methods, that enable multiple episodes to run in parallel while exploring the space of action bias terms in a principled manner. The use of batch BO in our target application domain is critical since real adaptive intervention studies must support multiple simultaneous participants in order to satisfy constraints on total study duration. To improve the BO exploration, we also investigate setting the TS prior parameters using a small micro-randomized trial (MRT).

We explore the above issues in the context of a recently proposed physical activity intervention simulation where the reward is in terms of step count and the actions correspond to the selection of different forms of contextualized motivational messages [Karine et al., 2023]. The simulation captures key aspects of the physical activity intervention domain including a habituation process affected by treatment volume and a disengagement process affected by context inference errors.

Our results show that optimizing action bias terms using batch BO and selecting actions using the proposed utility function leads to an extended Thompson sampler that outperform standard TS and full RL methods. Using local batch BO instead of global batch BO can further improve the performance. Moreover, modeling the reward variance in addition to the action bias terms can provide further mean performance improvements in some settings. These results suggest that our proposed approach has the potential to enhance the treatment efficacy of TS-based adaptive interventions.

**Our main contributions are:** (1) We introduce a novel extended Thompson sampling bandit model and learning algorithm that can solve a broader class of MDPs than standard Thompson sampling. (2) We show that the proposed extended Thompson sampler outperforms a range of RL baseline methods in the severely episode-limited setting. (3) We provide an implementation of the proposed approach. The code is available at: github.com/reml-lab/BOTS.

## 2 Methods

In this section, we describe our extension to Thompson sampling, our approach to optimizing the additional free parameters introduced, and the JITAI simulation environment. We describe the related work in Appendix A.1.

**Extended Thompson Sampling (xTS).** Our primary goal is to reduce the myopic nature of standard TS for contextual bandits when applied in the episodic MDP setting while maintaining low variance. Our proposed extended Thompson sampling approach is based on selecting actions according to a state-action utility that extends the linear Gaussian reward model used by TS as shown below.

$$u_{ta} = r_{ta} + \beta_a \tag{1}$$
$$p(r_{ta}|a, \mathbf{s}_t) = \mathcal{N}(r_{ta}; \theta_{ta}^\top \mathbf{s}_t, \sigma_{Ya}^2) \tag{2}$$
$$p(\theta_{ta}|\mu_{ta}, \Sigma_{ta}) = \mathcal{N}(\theta_{ta}; \mu_{ta}, \Sigma_{ta}) \tag{3}$$

where at each time $t$, $\mathbf{s}_t$ is the state (or context) vector, $r_{ta}$ is the reward for taking action $a$, $\theta_{ta}$ is a vector of weights for action $a$. We refer to the additional term $\beta_a$ included in the utility as the *action bias* for action $a$. The action bias values are fixed within each episode of xTS, but are optimized across episodes to maximize the total return of the resulting policy $\pi_{xTS}$. The policy $\pi_{xTS}$ is similar in

form to that of standard TS except that actions are selected according to the probability that they have the highest expected utility. The policy $\pi_{xTS}$ maintains the same posterior distribution over immediate rewards used by standard TS. We provide the pseudo code in Algorithm 2 in Appendix A.2.3.

The basic intuition for the xTS model is that the action bias parameters enable penalizing or promoting the selection of each action $a$ (regardless of the current state), based on action $a$'s average long term consequences effect on the total return. The inclusion of action bias parameters in the policy $\pi_{xTS}$ provides access to a strictly larger policy space than standard TS. At the same time, the number of additional parameters introduced by xTS is low, enabling the resulting approach to retain relatively low variance when learning from a limited number of episodes.

We note that in adaptive health interventions, it can be the case that the long term consequences of specific actions taken in different state have similar effect on the total return. For example, sending a message in a messaging-based adaptive health intervention will typically have a positive short-term impact, but this action will also increase habituation. Increased habituation then typically results in lower reward for some number of future time steps. Our proposed approach has the potential to result in an improved policy in such settings relative to TS, by recognizing that the long term consequences of actions on the total return may be mismatched with the expected immediate reward that they yield.

**Bayesian Optimization for xTS.** We now turn to the question of how to select the action bias values $\beta_a$ to optimize the xTS policy. Let $R = \sum_{t=1}^{T} r_t$ represent the total reward for an episode, where $r_t$ is the observed reward at time $t$, and $\mathbb{E}_\pi[R]$ represent the expected return when following policy $\pi$. Letting vectors $\beta = \{\beta_a\}_{0:A}$, $\mu_0 = \{\mu_{0a}\}_{0:A}$, $\Sigma_0 = \{\Sigma_{0a}\}_{0:A}$, and $\sigma_Y^2 = \{\sigma_{Ya}^2\}_{0:A}$, we propose to select $\beta$ to maximize the expected return of the policy $\pi_{xTS}(\beta, \mu_0, \Sigma_0, \sigma_Y^2)$:

$$\beta_* = \arg\max_\beta \mathbb{E}_{\pi_{xTS}(\beta, \mu_0, \Sigma_0, \sigma_Y^2)}[R] \tag{4}$$

To solve this optimization problem, we propose the use of batch Bayesian optimization. The target function is the expected return of the xTS policy with respect to the $\beta$ parameters. We begin by defining a schedule of batch sizes $B_i$ for rounds $i$ from 0 to $R$. On each round $i$, we use a batch acquisition function to select a batch of $B_i$ values of action bias parameters $\beta_{i1}, ..., \beta_{iB_i}$ based on the current Gaussian process approximation to the expected return function. We run one episode of xTS using the policy $\pi_{xTS}(\beta_{ib}, \mu_0, \Sigma_0, \sigma_Y^2)$ for $1 \leq b \leq B_i$ and obtain the corresponding returns $R_{ib}$. All episodes in round $i$ are run in parallel. When all episodes in round $i$ are complete, we use the $B_i$ pairs $(\beta_{ib}, R_{ib})$ to update the Gaussian process. We provide the pseudo code in Appendix Algorithm 1.

We perform experiments using several configurations of this algorithm. To start the BO procedure, we apply Sobol sampling to obtain an initial set of $\beta$ values. On later rounds, we typically apply the qEI acquisition function. When applying the acquisition function, we consider both unconstrained and local optimization. This choice corresponds to using global or local BO (e.g., TuRBO) [Balandat et al., 2020, Eriksson et al., 2019].

When using a global BO method to optimize xTS, this approach can represent optimal policies for any MDP where standard TS can represent an optimal policy, simply by learning to set $\beta_a = 0$ for all $a$. Moreover, our approach can also represent optimal policies for MDPs where the optimal action in each state corresponds to the action with the highest expected utility under some setting of the action bias parameters $\beta$. This is a strictly larger set of MDPs than can be solved using standard TS. However, it is clearly also strictly smaller than the set of all MDPs since the action bias terms are state independent, while the expected value of the next state in the optimal state-action value function $Q^*(\mathbf{s}, a)$ depends on the current state.

**Setting Thompson Sampling Parameters.**

As for typical TS, xTS requires that a prior $\mathcal{N}(\theta_a; \mu_{0a}, \Sigma_{0a})$ be specified. This prior can be set by hand using domain knowledge or hypothesized relationships between the immediate reward and the state variables. In the absence of strong domain knowledge, the prior can be set broadly to reflect this lack of prior knowledge. Finally, in the adaptive intervention domain, this prior is often set by applying Bayesian linear regression to a small number of episodes with randomized action selection referred to as a *micro-randomized trial* (MRT). Note that these episodes can also be performed in parallel.

When learning xTS over multiple rounds, we can fix the same reward model prior for all rounds, or update the prior from round to round using Bayesian linear regression. We call these strategies

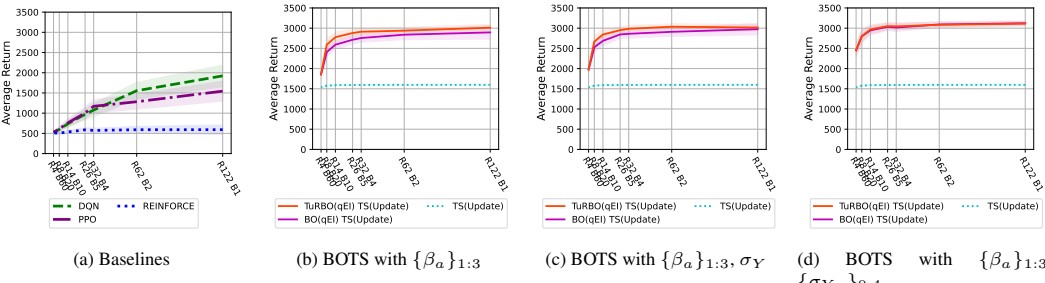

(a) Baselines         (b) BOTS with $\{\beta_a\}_{1:3}$     (c) BOTS with $\{\beta_a\}_{1:3}, \sigma_Y$     (d)  BOTS  with  $\{\beta_a\}_{1:3}$, $\{\sigma_{Ya}\}_{0:4}$

Figure 1: Results for severely episode-limited settings. Note: the x-axes show (number of rounds, batch size) combinations, not round index. In all experiments, BOTS shows a better performance in a low number of rounds.

TS(Fixed) and TS(Update) respectively. Lastly, we note that the reward variance $\sigma_Y^2$ terms also need to be chosen. These values can be chosen using domain knowledge, ascribed a hierarchical prior and marginalized over, or optimized over using the same procedure described above and in Appendix Algorithm 1, by augmenting the BO procedure to use $[\beta, \sigma_Y^2]$ as optimization variables.

**Adaptive Intervention Trial Simulation.** To evaluate our approach, we use the Just-in-Time Adaptive Intervention (JITAI) simulation environment introduced in [Karine et al., 2023]. The JITAI environment models a messaging-based physical activity intervention applied to a single individual. The JITAI environment simulates: a binary context variable ($C$), a real-valued habituation level ($H$), a real-valued disengagement risk ($D$), and the walking step count ($S$). We use step count $S$ as the reward in RL. The context $C$ can represent a behavioral state such as 'stressed'/ 'not stressed'. The simulation includes four actions: $a = 0$ indicates that no message is sent to the participant, $a = 1$ indicates that a non-contextualized message is sent, $a = 2$ indicates that a message customized to context 0 is sent, and $a = 3$ indicates that a message customized to context 1 is sent. The maximum study length is 50 days, with a daily intervention. We fix the action bias for action 0 to 0. The dynamics are such that sending any message causes habituation to increase, while sending an incorrectly contextualized message causes disengagement risk to increase. The effect of habituation is to decrease the ability of a message to positively influence the step count. When the disengagement risk exceeds a set threshold, the simulation models the participant withdrawing from the trial, eliminating all rewards after that time point, which is a common problem in mobile health studies. As a result of these dynamics, actions can strongly impact future rewards.

**BOTS overview.** We summarize and provide a graphical overview of the BOTS method, as applied in the JITAI setting, including setting TS priors using the MRT, in Appendix Figure 2.

## 3 Experiments and Results

We perform extensive experiments, which are detailed in Appendix B: basic MDPs, baselines without episode limits and the severely episode-limited settings. In severely episode-limited settings, we limit the total number of participants and thus episodes to 140. For BOTS, we allocate 10 individuals in rounds 0 and 1. This leaves a total of 120 from the budget of 140 participants. We evenly partition the remaining 120 across 2, 6, 12, 24, 30, 60 or 120 rounds. We run the same configurations on the baseline methods for comparison. We note that with 50 days required per round, the configurations that use 15 or more rounds would typically not be feasible in real studies, as they would require in excess of two years to run. In all our experiments, BOTS shows better performance in a low number of rounds, as shown in Figure 1.

## 4 Conclusions

Motivated by the use of RL methods to aid in the design of just-in-time adaptive health interventions, this paper focuses on practical methods that can deal with (1) severe constraints on the number of episodes available for policy learning, (2) constraints on the total duration of policy optimization studies, and (3) long terms consequences of actions. We have proposed a two-level approach that uses extended Thompson sampling (xTS) to select actions via an expected utility that includes fixed action bias terms, while a batch Bayesian optimization method is used to learn the action bias terms

over multiple rounds, to maximize the expected return of the xTS policy. We have also presented a practical method to set the linear reward model priors using a small micro-randomized trial. Our results show that under realistic constraints on the total episode count and the intervention study duration, the use of local batch Bayesian optimization combined with xTS outperforms the other methods considered and is a promising approach for deployment in real studies.

## Acknowledgements

This work is supported by National Institutes of Health National Cancer Institute, Office of Behavior and Social Sciences, and National Institute of Biomedical Imaging and Bioengineering through grants U01CA229445 and 1P41EB028242. The authors would like to thank Philip Thomas for helpful discussions related to this work.

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

# A Appendix

## A.1 Related work

Our overall approach is based on the use of local batch BO to learn the additional parameters of our proposes extended TS model. Our overall approach is closely related to the use of BO methods to tune hyper-parameters of machine learning algorithms [Snoek et al., 2012]. This includes the use of BO for direct policy search in hierarchical RL [Brochu et al., 2010]. Another recent work uses ensembles of bootstrapped neural networks coupled with randomized prior functions to approximate the TS posterior distribution [Osband et al., 2023]. This work proposes a modification of TS intended to enable solving any MDP, whereas we focus on a more modest extension of TS that retains low variance. The computational approach used by [Osband et al., 2023] is also quite different than our work which focuses on GP optimization of the additional terms introduced in the state-action utility function. Our approach is closely related to recent work on differentiable meta-learning of bandit policies [Boutilier et al., 2020] with the main difference being that we focus on the use of batch BO to explore many extended TS models simultaneously. Liao et al. [2022] consider the problem of learning corrections for linear TS, but their approach requires estimating an auxiliary MDP model and is less general than the method we propose. Trella et al. [2023] consider a similar model to the one considered in this paper, but use previously collected data with domain knowledge to hand-design an estimator for the correction terms in the context of a specific application.

## A.2 BOTS overview

In this section, we summarize the BOTS method described in Section 2, and provide a graphical overview of BOTS in the JITAI setting, and the BOTS pseudo codes.

### A.2.1 BOTS methods and parameters

We provide a summary of the BOTS methods and parameter space configurations in Tables 1 and 2.

Table 1: Summary of the BOTS methods

| Method | Type | Description |
| --- | --- | --- |
| BO(qEI) | global BO + extended TS | Batch BO with qEI acquisition function |
| TuRBO(qEI) | local BO + extended TS | Batch TuRBO with qEI acquisition function |

Table 2: Summary of the BOTS parameter space configuration

| Configuration | BO parameters | Description |
| --- | --- | --- |
| $\{\beta_a\}_{1:3}$ | $[\beta_1, \beta_2, \beta_3]$ | action bias terms (for actions 1, 2 and 3) |
| $\{\beta_a\}_{1:3}, \sigma_Y$ | $[\beta_1, \beta_2, \beta_3, \sigma_Y^2]$ | action bias terms, shared reward variance |
| $\{\beta_a\}_{1:3}, \{\sigma_{Ya}\}_{0:4}$ | $[\beta_1, \beta_2, \beta_3, \sigma_{Y_0}^2, \sigma_{Y_1}^2, \sigma_{Y_2}^2, \sigma_{Y_3}^2]$ | action bias terms, per-action reward variance |

### A.2.2 BOTS graphical overview in the JITAI setting

We provide a graphical overview of the BOTS method as applied in the JITAI setting, including setting TS priors using the MRT, in Figure 2. The BO parameters are summarized in Table 2.

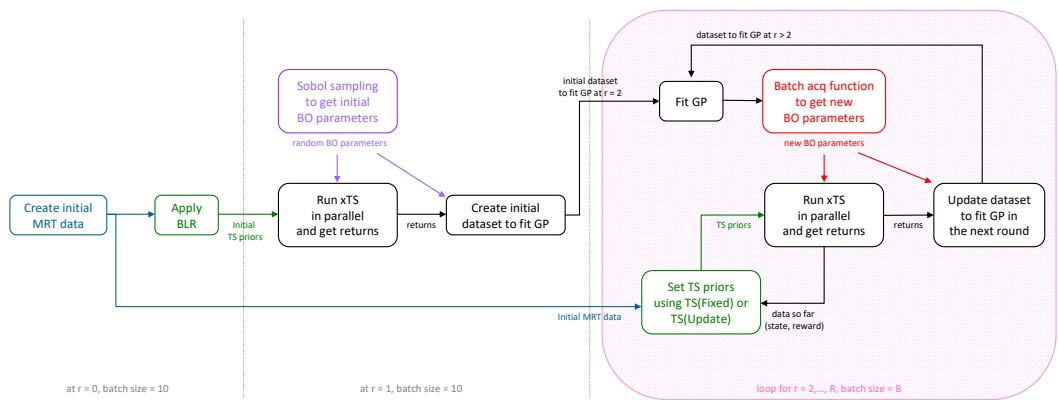

Figure 2: BOTS overview in the JITAI setting, including setting TS initial priors using an MRT. The TS priors are propagated across episodes using one of two strategies: TS(Fixed) where we fix the same reward model prior for all rounds, or TS(Update) where we update the prior from round to round. The BO parameter can be: vector $\{\beta_a\}_{1:3}$, vector $\{\beta_a\}_{1:3}, \sigma_Y$, or vector $\{\beta_a\}_{1:3}, \{\sigma_{Ya}\}_{0:4}$.

### A.2.3 BOTS algorithm

Below are the BOTS pseudo codes. The algorithm descriptions are provied in Section 2.

---

**Algorithm 1** BOTS: Batch Bayesian Optimization of Extended Thompson Sampling

---

Inputs $\{B_i\}_{0:R}, \mu_0, \Sigma_0, \sigma_Y^2$:
**for** $i = 0 : R$ **do**
    Use batch acq function to select $\beta_{ib}$ for $1 \leq b \leq B_i$
    **for all** $b = 1 : B_i$ **do in parallel**
        Run episode using policy $\pi_{xTS}(\beta_{ib}, \mu_0, \Sigma_0, \sigma_Y^2)$
        Obtain return $R_{ib}$
    **end for**
    Update GP using $\{(\beta_{ib}, R_{ib}) | 1 \leq b \leq B_i\}$
**end for**

---

**Algorithm 2** Extended Thompson Sampling policy $\pi_{xTS}$

---

Inputs $\{\beta_a\}_{0:A}, \{\mu_{0a}\}_{0:A}, \{\Sigma_{0a}\}_{0:A}, \{\sigma_{Ya}^2\}_{0:A}$
**for** t=0:T **do**
    Observe state $\mathbf{s}_t$
    **for** a=0:A **do**
        $\hat{\theta}_{ta} \sim \mathcal{N}(\theta_{ta}; \mu_{ta}, \Sigma_{ta})$
        $\hat{r}_{ta} \leftarrow \hat{\theta}_{ta}^\top \mathbf{s}_t$
        $\hat{u}_{ta} \leftarrow \hat{r}_{ta} + \beta_a$
    **end for**
    $a_t \leftarrow \arg\max_a \hat{u}_{ta}$
    Take action $a_t$. Observe $r_t$.
    **for** a=0:A **do**
        $\Sigma_{(t+1)a} \leftarrow \sigma_{Ya}^2 \left([a_t = a]\mathbf{s}_t^\top \mathbf{s}_t + \sigma_{Ya}^2 \Sigma_{ta}^{-1}\right)^{-1}$
        $\mu_{(t+1)a} \leftarrow \Sigma_{(t+1)a} \left([a_t = a](\sigma_{Ya}^2)^{-1} r_t \mathbf{s}_t + \Sigma_{ta}^{-1} \mu_{ta}\right)$
    **end for**
**end for**

---

### A.3 Implementation details

In this section, we provide the implementation details for the RL methods, basic MDPs, Bayesian Optimization (BO), and a summary of the JITAI environment specification.

### A.3.1 RL methods implementation details

As baseline methods, we consider REINFORCE and PPO as examples of policy gradient methods, and deep Q networks (DQN) as an example of a value function method. We also consider standard Thompson sampling (TS) with a fixed prior. We select the best hyper-parameters that maximize the performance, with the lowest number of episodes: the average return is around 3000 for the RL methods, and around 1500 for basic TS, using the JITAI setting described in Section 2. All experiments can be run on CPU, using Google Colab within 2GB of RAM.

**REINFORCE**. We use a one-layer policy network. We perform hyper-parameter search over hidden layer sizes $[32, 64, 128, 256]$, and Adam optimizer learning rates from 1e-6 to 1e-2. We report the results for 128 neurons, batch size $b = 64$, and Adam optimizer learning rate $lr = 6$e-4.

**DQN**. we use a two-layer policy network. We perform a hyper-parameter search over hidden layers sizes $[32, 64, 128, 256]$, batch sizes $[16, 32, 64]$, Adam optimizer learning rates from 1e-6 to 1e-2, and epsilon greedy exploration rate decrements from 1e-6 to 1e-3. We report the results for 128 neurons in each hidden layer, batch size $b = 64$, Adam optimizer learning rate $lr = 5$e-4, epsilon linear decrement $\delta_\epsilon = 0.001$, decaying $\epsilon$ from 1 to 0.01. The target Q network parameters are replaced every $K = 1000$ steps.

**PPO**. We use a two-layer policy network, and a three layers critic network. We perform a hyper-parameter search over hidden layers sizes $[32, 64, 128, 256]$, batch sizes $[16, 32, 64]$, Adam optimizer learning rates from 1e-6 to 1e-2, horizons from 10 to 40, policy clips from 0.1 to 0.5, and the other factors from .9 to 1.0. We report the results for 256 neurons in each hidden layer, batch size $b = 64$, Adam optimizer learning rate $lr = 0.001$, horizon $H = 20$, policy clip $c = 0.1$, discounted factor $\gamma = 0.99$ and Generalized Advantage Estimator (GAE) factor $\lambda = 0.95$.

**Q-Learning**. We perform a hyper-parameter search over learning rates from 0.01 to 0.1, discount factor $\gamma$ from 0.8 to 1., decaying $\epsilon$ from 1 to 0.01, with decay rates from 0.01 to 0.5. We report the results for learning rate $lr = 0.8$, $\gamma = 0.99$, and decay rate $\delta_\epsilon = 0.1$.

**TS(Fixed)**. We consider a baseline approach where TS is applied with a fixed prior. When fixing (e.g., not learning) elements of the Thompson sampling prior, we set them to the same values for all methods. We set $\mu_{0a} = 0$ for all $a$, and $\Sigma_{0a} = 100I$. We fix the Thompson sampling reward model noise variance to $\sigma^2_{Ya} = 25^2$ for all $a$.

**Batch RL agents**. We consider batch versions of the RL methods to match our focus on batch Bayesian optimization. In the case of REINFORCE, we simulate all episodes of a batch in parallel and compute an average policy gradient based on the average return of the batch elements. In the case of DQN, we again simulate all episodes in a batch in parallel. However, in this case we compute an average gradient across batch elements at each iteration of the DQN algorithm. Both REINFORCE and DQN are applicable in the full Markov decision process (MDP) setting with sequential dependence of rewards on prior state and actions.

### A.3.2 Basic MDPs implementation details

We implement three basic MDPs to empirically demonstrate scenarios in which basic TS and our proposed approach can and can not achieve optimal performance. These correspond to the MDPs in Figure 3. For the notation below: $s$ represents the current state value, $a$ is the action value given $s$, $s'$ is the next state value after taking action $a$, and $r$ is the reward.

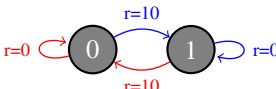

MDP1: TS ✓ BOTS ✓ Q ✓

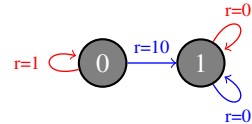

MDP2: TS ✗ BOTS ✓ Q ✓

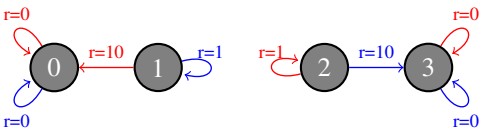

MDP3: TS ✗ BOTS ✗ Q ✓

Figure 3: Sketch of the three basic MDPs. Each MDP is annotated with an indication of whether each method considered finds the optimal policy (✓) or fail to find the optimal policy (✗).

**MDP1** has a binary state: $s \in \{0, 1\}$, and actions $a \in \{0, 1\}$. The episode length is 100. The transition function and reward function are both deterministic and are given below. The starting state is $s = 0$.

Table 3: MDP1 Transition Function

| State | Action | Next State |
|:-----:|:------:|:----------:|
| 0 | 0 | 0 |
| 0 | 1 | 1 |
| 1 | 0 | 0 |
| 1 | 1 | 1 |

Table 4: MDP1 Reward Function

| State | Action | Reward |
|:-----:|:------:|:------:|
| 0 | 0 | 0 |
| 0 | 1 | 10 |
| 1 | 0 | 10 |
| 1 | 1 | 0 |

**MDP2** has a binary state: $s \in \{0, 1\}$, and actions $a \in \{0, 1\}$. The episode length is 100. The transition function and reward function are both deterministic and are given below. The starting state is $s = 0$.

Table 5: MDP2 Transition Function

| State | Action | Next State |
|:-----:|:------:|:----------:|
| 0 | 0 | 0 |
| 0 | 1 | 1 |
| 1 | 0 | 1 |
| 1 | 1 | 1 |

Table 6: MDP2 Reward Function

| State | Action | Reward |
|:-----:|:------:|:------:|
| 0 | 0 | 1 |
| 0 | 1 | 10 |
| 1 | 0 | 0 |
| 1 | 1 | 0 |

**MDP3** has a state with four possible values: $s \in \{0, 1, 2, 3\}$, and actions $a \in \{0, 1\}$. The episode length is 100. The transition function and reward function are both deterministic and are given below. The starting state is chosen randomly with $s = 1$ or 2.

Table 7: MDP3 Transition Function

| State | Action | Next State |
|:-----:|:------:|:----------:|
| 0 | 0 | 0 |
| 0 | 1 | 0 |
| 1 | 0 | 0 |
| 1 | 1 | 1 |
| 2 | 0 | 2 |
| 2 | 1 | 3 |
| 3 | 0 | 3 |
| 3 | 1 | 3 |

Table 8: MDP3 Reward Function

| State | Action | Reward |
|:-----:|:------:|:------:|
| 0 | 0 | 0 |
| 0 | 1 | 0 |
| 1 | 0 | 10 |
| 1 | 1 | 1 |
| 2 | 0 | 1 |
| 2 | 1 | 10 |
| 3 | 0 | 0 |
| 3 | 1 | 0 |

### A.3.3 Summary of JITAI simulation environment specifications

The JITAI simulation environment introduced in [Karine et al., 2023], is a behavioral simulation environment that mimics a participant's behaviors in a mobile health study, where the interventions (actions) are the messages sent to the participant. We summarize the JITAI environment specifications in Tables 9 and 10.

Table 9: Possible action values

| Action value | Description |
|---|---|
| $a = 0$ | No message is sent to the participant. |
| $a = 1$ | A non-contextualized message is sent to the participant. |
| $a = 2$ | A message customized to context 0 is sent to the participant. |
| $a = 3$ | A message customized to context 1 is sent to the participant. |

Table 10: JITAI simulation variables

| Variable | Description | Values |
|---|---|---|
| $c_t$ | true context | $\{0, 1\}$ |
| $\mathbf{p}_t$ | context probabilities | $\Delta^1$ |
| $l_t$ | inferred context | $\{0,1\}$ |
| $d_t$ | disengagement risk level | $[0, 1]$ |
| $h_t$ | habituation level | $[0, 1]$ |
| $s_t$ | step count | $\mathbb{N}$ |

We also provide a summary of the **deterministic dynamics** below.

$$c_t \sim Ber(0.5)$$
$$x_t \sim \mathcal{N}(c_t, \sigma^2)$$
$$\mathbf{p}_t = [p_{0t}, p_{1t}]$$
$$l_t = \arg\max_{c \in \{0,1\}} p_{ct}$$
$$h_{t+1} = \begin{cases} (1 - \delta_h) \cdot h_t & \text{if } a_t = 0 \\ \min(1, h_t + \epsilon_h) & \text{otherwise} \end{cases}$$
$$d_{t+1} = \begin{cases} d_t & \text{if } a_t = 0 \\ (1 - \delta_d) \cdot d_t & \text{if } a_t = 1 \text{ or } a_t = c_t + 2 \\ \min(1, d_t + \epsilon_d) & \text{otherwise} \end{cases}$$
$$s_{t+1} = \begin{cases} \mu_s + (1 - h_{t+1}) \cdot \rho_1 & \text{if } a_t = 1 \\ \mu_s + (1 - h_{t+1}) \cdot \rho_2 & \text{if } a_t = c_t + 2 \\ \mu_s & \text{otherwise} \end{cases}$$

where $c_t$ is the true context, $x_t$ is the true context feature, $\sigma$ is the context uncertainty, $\mathbf{p}_t$ is a vector of context probabilities, $p_{0t}$ is the probability of context 0, $p_{1t}$ is the probability of context 1 (where $p_{1t} = 1 - p_{0t}$), $l_t$ is the inferred context, $h_t$ is the habituation level, $d_t$ is the disengagement risk, $s_t$ is the step count ($s_t$ is the participant's number of walking steps), and $a_t$ is the action value at time $t$.

The behavioral dynamics can be tuned using the hyperparameters: disengagement risk decay $\delta_d$, disengagement risk increment $\epsilon_d$, habituation decay $\delta_h$, and habituation increment $\epsilon_h$.

The default hyperparameters values for the base JITAI simulation environment are: $\delta_h = 0.1$, $\epsilon_h = 0.05$, $\delta_d = 0.1$, $\epsilon_d = 0.4$, $\mu_s = [0.1, 0.1]$, $\rho_1 = 50$, $\rho_2 = 200$, disengagement threshold $D_{threshold} = 0.99$ (the study ends if $d_t$ exceeds $D_{threshold}$). The maximum study length is 50 days with one intervention per day. The context uncertainty $\sigma$ is typically set by the user, with value $\sigma \in [0.1, 2]$, as detailed in [Karine et al., 2023].

### A.3.4 BO implementation details

To apply Bayesian optimization, we need to define bounds on all parameters. For the action bias terms, we use $-100 \leq \beta_a \leq 0$ for $a > 0$ as we select actions to have negative or neutral impacts in our environment. For the reward variances, we use $0.1 \leq \sigma_Y \leq 50^2$ and $0.1 \leq \sigma_{Ya} \leq 50^2$. When applying Sobol sampling, we sample within these bounds.

The GP is fit to the normalized BO parameters and standardized returns. We initialize the GP likelihood noise standard deviation to $\sigma_\epsilon = 0.7$. We place a hierarchical prior on the Matérn 5/2 kernel parameters including a Gamma(3.0, 6.0) prior on the lengthscale $\sigma_l$ and a Gamma(2.0, 0.15) prior on the output scale $\sigma_f$. We use marginal likelihood maximization to refine the kernel hyper-parameters during BO. We use the BoTorch implementations of the qEI acquisition function and the TuRBO method.

## B  Experiments and Results

In this section, we describe the experiments and results for basic MDPs and for JITAI environment.

### B.1  Basic MDPs Experiments

We show results on basic MDPs to empirically demonstrate scenarios in which basic TS and our proposed approach can and can not achieve optimal performance. We use three MDPs as shown in Figure 3. All three MDPs are deterministic and have two actions (red and blue). The state transitions are deterministic and are shown by arrows. The arrows are annotated with the immediate rewards. The rewards are also deterministic. MDP1 and MDP2 consist of a binary state and a binary action, with starting state $s_1 = 0$. MDP3 consists of a state with values $[0, 1, 2, 3]$, and a binary action, where the starting state is chosen uniformly at random from the set $\{1, 2\}$. The episode length is $100$. We provide the implementation details for these MDPs in Appendix A.3.2.

We consider the application of three methods. First, we apply classical tabular Q-Learning with an epsilon greedy exploration policy, which will converge to the optimal policy for these MDPs. Details are provided in Appendix A.3.1. We also apply standard TS starting from an uniformed prior and applied in a mode where the posterior obtained at the end of one episode becomes the prior for the next episode. Finally, we apply BOTS starting from the same prior as standard TS. We use TuRBO as the BO approach. On the first BOTS round, we use Sobol sampling with a batch size of 2. For the remaining rounds, we use one episode per batch and apply the EI acquisition function. Figure 4 shows average return versus the number of episodes results for each method.

MDP1 corresponds to a case where the action $a$ with the highest immediate reward in each state $s$ also corresponds to the action with highest value of $Q^*(s, a)$. We can see that all three methods converge to the same optimal return. MDP2 corresponds to a case where the action $a$ with the highest immediate reward in each state $s$ does not correspond to the action with highest value of $Q^*(s, a)$, but there do exist settings of the action bias parameters such that the action with the highest value of the BOTS utility function corresponds to the action with the highest value of $Q^*(s, a)$. For this MDP, TS has poor performance while BOTS matches the performance of Q-learning. Lastly, MDP3 is a case where the BOTS utility function and action bias parameters are not sufficient to learn an optimal policy. For this MDP, both TS and BOTS fail to match Q-Learning, as expected. The results are shown in Figure 4.

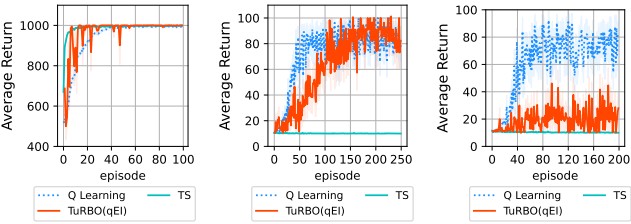

Figure 4: Results for MDP1 (left), MDP2 (middle) and MDP3 (right).

## B.2 JITAI Simulation Experiments

In this section, we describe the experiments using the JITAI simulation environment. We start with the simulator settings, then describe performance metrics, method configurations, and the results.

**JITAI Simulation Environment Configuration.** To model a realistic total number of participants in multi-round mobile health adaptive intervention studies, we set 140 as the total number of simulated participants. Each participant in an intervention trial corresponds to one RL episode. The maximum episode length is $L = 50$ days. The JITAI simulation environment contains a number of parameters for tuning the simulation dynamics [Karine et al., 2023]. We report results using context uncertainty $\sigma = 0.1$, disengagement threshold $D = 0.99$, habituation decay $\delta_h = 0.1$, habituation increment $\epsilon_h = 0.05$, disengagement decay $\delta_d = 0.1$, disengagement increment $\epsilon_d = 0.4$, base rewards for non-tailored and tailored message $\rho_1 = 50$ and $\rho_2 = 200$ respectively.

**Performance Metrics.** In our specific adaptive health intervention setting, the efficacy of the treatment received by each individual during the course of the trial is an important performance metric. We thus focus on the average return over all participants in the adaptive intervention optimization study as the primary performance metric in our experiments. We report results in terms of the mean of the average return computed over the 10 simulated repetitions of the study. We also report the standard error computed over the average return of the 10 repetitions of the study. The standard errors are presented graphically as shaded regions.

**Method Configurations.** In terms of baselines, we consider REINFORCE and PPO as examples of policy gradient methods, and deep Q networks (DQN) as an example of a value function method. We implement batch versions of these methods to match our focus on batch BO. The implementation details are in Appendix A.3.1.

For BOTS, we consider an application of the method where we simulate an MRT with 10 individuals in parallel to set the reward model priors. We then perform a Sobol sampling round to initialize the GP with 10 individuals in parallel. We then perform a number of BO rounds that differs per experiment. We use the qEI acquisition function with the BO variants: global batch BO, which we denote by BO(qEI), and trust region batch BO, which we denote by TuRBO(qEI). As with standard TS, we can either fix the same reward model prior for all rounds, or update the prior from round to round. We provide implementation details in Appendix A.3.4, and a graphical overview in Figure 2.

**Performance of baselines without episode limits.** In this experiment we assess the performance limits of baseline methods on the simulation environment in the absence of constraints on the episode count. We perform the experiments using REINFORCE, DQN, PPO, and standard TS with fixed and updated priors. We run 10 repetitions of 1500 episodes and report average results in Figure 5. The results show that REINFORCE, DQN and PPO require a large number of sequential episodes to reach high performance: about 1500 episodes for REINFORCE (e.g., up to 75,000 days) and more than 100 episodes for DQN and PPO (e.g., up to 5,000 days). As expected, the standard TS methods show fast convergence, but to lower performance than the full RL methods. These results motivate the need for BOTS in terms of an approach that can both enable parallelization across episodes to reduce total study time while achieving a performance improvement relative to standard TS. We provide the results for the experiments without episode limits in Figure 5.

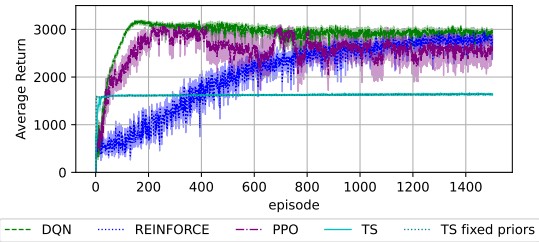

Figure 5: Baseline results without episode limits.