# OpenReview forum: "BOTS: Batch Bayesian Optimization of Extended Thompson Sampling for Severely Episode-Limited RL Settings"
_NeurIPS.cc/2024/Workshop/BDU — NeurIPS BDU Workshop 2024 Poster_

### Official Review · Reviewer_aRqE · 2024-10-05
**Review of the Paper on Extending Linear Thompson Sampling Bandits**

**Rating:** 8
**Confidence:** 4

**Review:**

This paper presents a compelling advancement in the application of reinforcement learning (RL), specifically addressing the challenges faced in real-world trials where episodes are limited by practical constraints. The authors have effectively extended the linear Thompson sampling bandit by incorporating a state-action utility function, which includes both the estimated immediate reward and an action bias term. This innovative approach is particularly noteworthy for several reasons:

Methodology: The integration of batch Bayesian optimization to learn action bias terms is a novel approach that enhances the effectiveness of Thompson sampling. By maximizing the expected return, the proposed method broadens the applicability to a wider class of Markov decision processes (MDPs).
Practical Relevance: The focus on optimizing adaptive health interventions highlights the real-world applicability of this research. The ability to achieve significant improvements in total return with fewer episodes compared to standard methods is a major advantage in settings where resources are limited.
Validation: The use of an adaptive intervention simulation environment to demonstrate the superiority of the proposed method adds credibility to the findings. The clear outperformance over standard Thompson sampling, as well as traditional value function and policy gradient methods, is well-substantiated by the results.
Contribution: By addressing the bias-variance trade-off in contextual bandit methods, this paper contributes valuable insights that could influence future research in RL and its applications in various domains.

Overall, this paper makes a significant contribution to the field of reinforcement learning, offering a practical and effective solution to a common limitation in real-world applications. The authors have demonstrated both technical rigor and practical impact, making this work a valuable addition to the body of RL literature.

---

### Official Review · Reviewer_SxFs · 2024-10-06
**Review of BOTS for extended Thompson sampling in episode limited settings**

**Rating:** 5
**Confidence:** 4

**Review:**

# Summary
The paper studies Thompson sampling in severely episode restricted settings. The authors extend the traditional linear TS bandit algorithm by including a state-action utility function that includes an action bias term learned across episodes. Such an approach relies on Bayesian optimization applied to the expected return of the (extended) Thompson sampler.

The main approach that I understood is that they add a beta_a term to the reward term (r_{t,a} see Equation 1) for optimization. The beta_a term indicates a "long term reward", or mean reward over states. While this is a good heuristic as shown by the paper's experiments why this heuristic was chosen (or why it works) is not clear.

---

# Strengths
- The notations are very clear and elucidate the method as required.
- The paper proposes methods to run the batch settings in parallel, which may be a computational advantage.
- The paper shows good emperical results over the settings considered.

---

# Weaknesses
- Are there theoretical guarantees on BOTS performance?
- The section from lines 324 to 332 may require a rewrite. There are too many undefined terms, or terms that are defined in other papers, which are referenced here without explanation.
- While I understand that this is not a theory oriented paper, do the authors have an intuition as to why the "action bias term" (which seems to me to be a heuristic) has a positive effect on the algorithm performance? I believe there may be counterexamples where the state has a large effect on which action should be chosen, thereby making the heuristic redundant for that scenario. Since the authors are working with simulations, what kind of settings are taken up in those simulations (where this method yields good results)?

---

# Questions for authors:
- What is the motivation for using a Matérn 5/2 covariance function (Equation 8 in page 9 of the appendix)?
- What is the main contribution of the paper? Is it that you use a state-action utility function, or is it that the state-action utility function contains an action bias term. Has such a utility function been explored in literature before?
- In figure 1, why do the baselines start at a return of 500, whereas the BOTS versions start at close to 2000, at the extreme left end of the plot?

---

# Comments, Suggestions, Typos:
- No link to code has been provided (see line 65).

---

I liked the paper, though there are some missing parts. May increase score depending on the answers to the questions above.

---

### Official Review · Reviewer_Th23 · 2024-10-06
**An interesting approach. The paper's readability and clarity could be improved.**

**Rating:** 6
**Confidence:** 2

**Review:**

In this paper, the authors propose introducing a set of “action bias” terms in Thomposon sampling-based approaches for solving contextual bandit problems, and combine this with batch BO, arguing this helps with solving episode-limited RL problems. The authors demonstrate significant improvements in performance compared to a number of baselines.

Overall, this is an interesting method, however it was not fully clear to me what the assumptions of the setting are. For example, equations 1 to 3 mix together terms that define the environment (e.g. the reward distribution) and terms that are introduced by the method (e.g. the bias term). A clear statement of the contextual bandits problem, together with the aspects that are specific to their algorithm would be beneficial for the readability of the paper.

It would be good to see the performance of a random sampling baseline, in addition to the baselines in figure (a).

L23: “human subjects trial”: perhaps there is a typo here.

---

### Official Review · Reviewer_U1zN · 2024-10-07
**Review comments for “BOTS: Batch Bayesian Optimization of Extended Thompson Sampling for Severely Episode-Limited RL Settings”**

**Rating:** 6
**Confidence:** 3

**Review:**

This paper proposed an extended Thompson Sampling method (xTS), designed to address the myopic issues of traditional Thompson Sampling (TS) in episodic Markov Decision Processes (MDP) while maintaining the advantage of low variance. The method introduced action bias terms into the state-action utility function and utilized batch Bayesian Optimization (BO) to optimize these biases to enhance the overall expected return of the policy. Additionally, the paper also described the JITAI simulation environment, which simulated a message-based physical activity intervention, exploring how action bias terms could potentially optimize the effectiveness of health interventions.

Specific advantages and disadvantages are shown below：

Pros：
1. By introducing action biases and utilizing batch Bayesian Optimization to learn these biases, the paper provided a new solution for improving decision-making under model-based uncertainty and data-scarce environments.

2. The authors designed various scenarios to test different reinforcement learning methods, including classical Q-learning, Thompson Sampling, and the newly proposed BOTS method, conducting comprehensive comparative experiments. Additionally, the paper provided detailed experimental results, including performance comparisons under various parameter settings and contrasts between stochastic and fixed dynamics, lending strong credibility to the findings.

3. The paper successfully combined theoretical methods with specific health intervention scenarios, particularly by optimizing long-term health intervention decision processes through enhanced Thompson Sampling and batch Bayesian optimization methods.

Cons：

1. Is the computational overhead taken into account? Although the BOTS method performed well, the paper did not sufficiently discuss its computational complexity and the overhead involved in practical applications. The demands for computational resources and real-time processing might have limited its practicality in broader applications.

2. Although the paper provided detailed descriptions of the experimental setup, it lacked an in-depth analysis of the sensitivity of key parameters, such as the specific impact of stochastic parameters on model performance. While the method included several tunable parameters to optimize performance, the paper did not sufficiently explore the sensitivity of these parameters or their optimal configuration.

3. The research has focused primarily on the JITAI environment, which may limit the generalizability of the method for validation. Is it equally valid for other environments?

Overall, the paper demonstrated excellent achievements in theoretical innovation and method application. It also provided a detailed theoretical background, as well as specific descriptions of the experimental process and result analysis. However, there may still be room for improvement in terms of computational complexity and parameter sensitivity analysis, and further exploration is needed for broader validation in different environments. Finally, I hope the authors can provide open-source code and further improvements.

---

### Decision · Program_Chairs · 2024-10-09

Accept (Poster)